# *Hymenoptera* Venom Immunotherapy: Immune Mechanisms of Induced Protection and Tolerance

**DOI:** 10.3390/cells10071575

**Published:** 2021-06-22

**Authors:** Ajda Demšar Luzar, Peter Korošec, Mitja Košnik, Mihaela Zidarn, Matija Rijavec

**Affiliations:** 1University Clinic of Respiratory and Allergic Diseases Golnik, 4204 Golnik, Slovenia; ajda.demsarluzar@klinika-golnik.si (A.D.L.); peter.korosec@klinika-golnik.si (P.K.); mitja.kosnik@klinika-golnik.si (M.K.); mihaela.zidarn@klinika-golnik.si (M.Z.); 2Faculty of Medicine, University of Ljubljana, 1000 Ljubljana, Slovenia; 3Biotechnical Faculty, University of Ljubljana, 1000 Ljubljana, Slovenia

**Keywords:** *Hymenoptera* venom immunotherapy, immune mechanisms, short-term protection, long-term tolerance

## Abstract

*Hymenoptera* venom allergy is one of the most severe allergic diseases, with a considerable prevalence of anaphylactic reaction, making it potentially lethal. In this review, we provide an overview of the current knowledge and recent findings in understanding induced immune mechanisms during different phases of venom immunotherapy. We focus on protection mechanisms that occur early, during the build-up phase, and on the immune tolerance, which occurs later, during and after *Hymenoptera* venom immunotherapy. The short-term protection seems to be established by the early desensitization of mast cells and basophils, which plays a crucial role in preventing anaphylaxis during the build-up phase of treatment. The early generation of blocking IgG antibodies seems to be one of the main reasons for the lower activation of effector cells. Long-term tolerance is reached after at least three years of venom immunotherapy. A decrease in basophil responsiveness correlates with tolerated sting challenge. Furthermore, the persistent decline in IgE levels and, by monitoring the cytokine profiles, a shift from a Th2 to Th1 immune response, can be observed. In addition, the generation of regulatory T and B cells has proven to be essential for inducing allergen tolerance. Most studies on the mechanisms and effectiveness data have been obtained during venom immunotherapy (VIT). Despite the high success rate of VIT, allergen tolerance may not persist for a prolonged time. There is not much known about immune mechanisms that assure long-term tolerance post-therapy.

## 1. Introduction

The immune system has the capacity of protecting the organism from pathogens by differentiating between foreign and self-components, thereby obtaining a state of self-tolerance. Allergic reactions arise because of the dysregulation of the immune system [1].

*Hymenoptera* venom allergy (HVA) is an IgE-mediated allergic disease caused by cross-linking receptor-bound IgE antibodies on the surface of mast cells and basophils [2,3]. The clinical picture varies from large local reactions (LLR) at the sting site to systemic reactions (SRs). A large local reaction is swelling larger than 10 cm in diameter that lasts longer than 24 h [4]. SRs vary greatly in severity, from moderate reactions consisting of generalized skin symptoms, to severe life-threatening anaphylactic reactions affecting the cardiac and respiratory system [5]. The prevalence of systemic reactions is 0.3–8.9%, with anaphylaxis in 0.3–42.8% of cases [6]. For patients with anaphylactic reactions to *Hymenoptera* venom, the only disease-modifying treatment is allergen-specific venom immunotherapy (VIT) [7].

## 2. Venom Immunotherapy

Immunotherapy aims to restore immune tolerance and thus eliminate systemic allergic reactions after insects’ stings [7]. The first immunotherapy using pure venom extract was carried out in 1974 [8]. Since then, many improvements have been made. Venom immunotherapy is a procedure in which insect venom preparations are administered as a series of subcutaneous injections. It is a two-step procedure consisting of the build-up phase and the maintenance phase [7]. The time to reach the maintenance dose depends on the protocol used—namely, conventional, rush, ultra-rush, or cluster protocol. The build-up phase can take several weeks or months in conventional protocols [9], or only a few days or hours in rush or ultra-rush protocols [9,10]. Cluster protocol represents an alternative regimen for conventional protocols. The recommended starting dose is between 0.001 and 0.1 µg. Subcutaneous injections in the maintenance phase are usually given in four-week intervals in the first year of treatment, every six weeks in the second year of treatment, and every eight weeks from the third to the fifth year of VIT [11]. A maintenance dose of 100 µg is used in the majority of patients. In patients with SRs after a field sting or sting challenge while on 100 µg of maintenance dose, upping the dose to 200 µg is recommended [7]. The detailed scheme of the particular protocol is shown in Table 1 [7,12]. The protocol used may be adapted individually, depending on patients’ reactivity.

In general, *Hymenoptera* VIT is considered to be safe, although in some cases potentially life-threatening SRs can occur. It has been suggested that rush/ultra-rush protocols can result in a higher rate of SRs compared with cluster or conventional protocols. However, the study data addressing this issue are conflicting [13]. In the latest study, Pospischil et al. demonstrated that accelerated VIT protocols, namely rush and ultra-rush, are safer than cluster protocols, as they displayed fewer SRs [14].

After the introduction of VIT in the early 1970s, it was believed that lifelong therapy would be necessary. Later, it was demonstrated that VIT can be safely stopped in the majority of the patients after three to five years of treatment. VIT has been shown to be effective in 77–84% of patients treated with honeybee venom and in 91–96% of patients treated with vespid venom [15,16]. In contrast to aero and food immunotherapies, the immune tolerance established during VIT is considered to be lifelong, even after the discontinuation of treatment [17,18,19]. Although many studies have focused on the detection of insect venom sensitization, no biomarker for therapy monitoring and evaluation of the efficiency has been established. To this day, a controlled sting challenge is the golden standard for the evaluation of venom tolerance, indicating clinical protection [7,20]. However, in selected cases, VIT should not be discontinued. The general consensus is that patients with an initial severe sting reaction and patients diagnosed with clonal mast cell disorder (often associated with elevated basal serum tryptase and *KIT* D816V mutation) should receive lifelong VIT [21,22]. Even though it is well documented that VIT provokes venom tolerance in the majority of treated patients, the exact underlying mechanism remains unclear.

## 3. Immune Mechanisms during Venom Immunotherapy

For understanding the induced immune mechanisms that occur during VIT, it is essential to know how HVA is manifested (Figure 1). After first exposure, in the initial phase of sensitization to venom, priming of T helper type 2 cells (Th2) takes place, resulting in the production of interleukin-4 (IL-4) and interleukin-13 (IL-13). The signaling pathway triggers immunoglobulin-E (IgE) production by B cells. Upon re-exposure to venom, IgE/high-affinity IgE receptor (FcεRI) cross-linking on the surface of mast cells, basophils, and antigen-presenting cells trigger the release of preformed mediators of inflammation, such as histamine, leukotrienes, prostaglandins, tryptase, and cytokines. This results in the development of a type I hypersensitivity reaction [24,25,26].

Treatment of HVA with VIT can reverse the hypersensitivity, and has an impact on antigen recognition, making it not detrimental to health. Low-dose repeated exposure to an allergen leads to limited or no inflammation. As a result, the differentiation and activation of naïve T cells are shifted towards the Th1 and regulatory T cells (Tregs) response, which subsequently modifies the B cell response, resulting in an increase of IgG4 antibodies. When allergen re-exposure happens, IgG4 and possibly other factors compete with IgE and inhibit IgE-mediated degranulation of the effector cells. To summarize, existing memory is rebalanced in both T cell and B cell compartments [2,27,28].

Acquired allergen tolerance is characterized by several mechanisms. During the build-up phase, a non-specific short-term protection effect can be observed, while in the maintenance phase of VIT, the induction of specific long-term tolerance mechanisms takes place (Figure 2). Different studies, described in the following paragraphs, have been made, suggesting the mechanisms and cells involved.

## 4. Mechanisms of Short-Term Protection

It is well known that for long-lasting allergen tolerance, at least three to five years of VIT are needed, even though it has been confirmed that early clinical protection is established after reaching the maintenance dose (MD), namely a short-term protection effect [29]. Our knowledge of short-term protection established during the first days of VIT is incomplete to explain the protective and tolerogenic pathways; however, several protective mechanisms have been proposed.

### 4.1. Basophils and Mast Cells

Mast cells and basophils are crucial effector cells in the inflammatory response in allergies. Both share functional similarities, such as the expression of the FcεRI on their surface, are derived from the same haematopoietic precursor, and release inflammatory mediators such as histamine, leukotrienes, and prostaglandins [30].

The early response to VIT is characterized by the desensitization of mast cells and basophils, resulting in a decrease in degranulation, lower activation status, and reduced production of IL-4 and IL-13 [31]. Even though allergen-specific IgE levels increase during the build-up phase, desensitization of mast cells and basophils can be observed. The desensitization effect can be explained by the rapid upregulation of histamine receptor 2 (H2R), which was observed within the first 6 hours of the build-up phase [32]. One of the main soluble factors released from activated mast cells and basophils is histamine. Histamine mediates its effects through histamine receptors. As H2R is associated with the tolerogenic immune response, it might contribute to the immunosilencing of effector cells in the early phase of VIT [32,33]. Lower effector cell responsiveness is essential to prevent systemic anaphylaxis. Another proposed mechanism of clinically induced basophil desensitization is the reduction of FcεRI expression [34]. FcεRI plays a central role in regulating the signal transduction of the IgE-mediated response. These receptors are expressed on the surface of mast cells and basophils [35]. FcεRI mediated basophil desensitization may have an important role in achieving an early protective state [36].

It has also been proposed that piecemeal release of histamine and leukotrienes below the threshold of systemic anaphylaxis might decrease the granule content of mediators, and subsequently, it might affect the threshold of activation of mast cells and basophils [37].

Several studies have reported a decrease in the blood basophil count, which occurs only during the build-up phase of VIT, returning to baseline values at the time of the first maintenance dose (approximately after 1 week of VIT) [31,34]. It is essential to mention that the effect of VIT on basophils and mast cells during the build-up phase seems to be venom-non specific [38].

### 4.2. Regulatory Cells of the Immune System

Maintenance of peripheral tolerance to antigens, including allergens, requires a fine balance between regulatory and effector T cells. Tregs represent a heterogeneous group of T-cell subsets. Briefly, naïve CD3+CD4+ T cells can be classified as thymus-derived CD25+FOXP3+ Tregs or peripherally derived Tregs generated outside the thymus. Peripheral Tregs can be sorted as peripherally induced FOXP3+ T cells, IL-10 producing Tregs (Tr1), and TGF-β producing Th3 cells [39]. Studies suggest that thymus-derived Tregs are specific for self-peptides, whereas peripherally induced Tregs are required to avoid pathologies triggered by antigens [40]. Tregs possess the suppressive capacity acting at different levels of immune mechanisms. There are four main mechanisms of anti-inflammatory activity of Tregs, as follows: (i) suppressive cytokines (IL-10, TGF-β, and IL-35), (ii) metabolic disruption mechanisms, (iii) suppression of dendritic cell (DC) activation by membrane-bound molecules, and (iv) cytolysis [39,41].

The peripheral activity measured as the number of Tregs subset defined as CD4+CD25+ does not show significant dynamics of change during the build-up phase of VIT [42]. Furthermore, the early changes observed on the mRNA level of peripheral blood mononuclear cells (PBMC) included significantly upregulated the degradation of tryptophan. Tryptophan degradation is directly linked to the suppression of T cell responses. The mechanism described is one of the earliest pro-tolerogenic mechanisms detected only a few hours after the first allergen injection [43].

### 4.3. Other Immune Cells

The selective apoptosis of cell sub fractions represents another way of possible tolerance induction. At as early as day three of the build-up phase of VIT, increased apoptosis of monocytes was reported by Bussman et al. [43]. Monocytes have the ability to sense the environment further differentiating into macrophages and DCs, which are able to present antigens and induce T-cell activation. Apoptosis of monocytes thus has an immunosuppressive effect on T-cells.

Upregulation of the protein expression of antigen-presenting cells (APCs) inhibitory receptors immunoglobulin-like transcript 3 (ILT3) and 4 (ILT4) was observed in the initial phase of VIT. Promoting the upregulation of tolerogenic surface markers such as ILT3 and ILT4 negatively affects the activation of APCs, thus acting as immunosuppressive on T-cells [43].

### 4.4. Antibodies

Among the mechanisms proposed to account for VIT efficacy are also changes in the antibody levels and their activity. The aforementioned successful desensitization to venom during the early stages of VIT may be a result of increased allergen-specific blocking IgA, IgG1, and IgG4 antibodies [44].

Blocking antibodies can compete with specific-IgE (sIgE) for allergens, and thus prevent IgE/allergen interaction, further preventing the cross-linking of FcεRI, therefore inhibiting the degranulation of mast cells and basophils [45]. Blocking IgG4 antibodies also stop the allergen-induced memory IgE production by blocking low-affinity receptors (FcγRIIb) on B cells, resulting in the inhibition of the IgE-facilitated presentation of allergens to T cells [46]. Studies suggest that the level of IgG is of value for predicting the risk of a systemic reaction after a sting. After 4 years of VIT, the predictive value of IgG4 declines, suggesting a long-term tolerance mechanism independent of venom-specific IgG [47].

### 4.5. Cytokines

Cytokines are small, secreted proteins with the ability to regulate the immune response. One of the most important indicators of induced immune tolerance is cytokine IL-10. Bussmann et al. observed a significant upregulation of IL-10 serum levels from day three of the build-up phase of VIT [43]. IL-10 is a major regulatory cytokine of inflammatory responses secreted by the Tregs subset, with a tolerogenic effect on the immune system [48]. Furthermore, the gradual reduction of IL-4 can be observed as early as 24 h from the initiation of VIT. Lower levels of IL-4 suggest reduced activation of the Th2 response [49].

## 5. Mechanisms of Induced Long-Term Tolerance

The duration of VIT needed to achieve long-term tolerance varies among patients. Termination after one or two years leads to a relapse rate of 22–27% [50], while some studies suggest three years should be sufficient for patients with mild to moderate initial sting reactions [51]. Nevertheless, a minimum of five years is suggested to be optimal to induce long-term tolerance [52,53].

As the majority of clinical effects of induced *Hymenoptera* venom tolerance achieved during VIT have still not been completely explained immunologically, there are no available biological markers that could predict the efficiency of VIT. Looking from that perspective, it is clearly important to deepen our knowledge of the immunological mechanisms that occur during VIT.

### 5.1. Basophils and Mast Cells

Some of the late effects observed during VIT are a decrease in tissue infiltration and the mediator release of mast cells and basophils. Many studies have been published concerning VIT monitoring using basophil activation test (BAT), and it has been observed that the tolerated sting challenge correlates with a decrease in basophil responsiveness and hence decreased basophil activation thresholds [36,54,55,56]. Basophil activation has been demonstrated to be inhibited by various subclasses of IgG antibodies that are produced during VIT. IgG antibodies suppress FcεRI activity on basophils and are related to low-affinity IgG receptors (FcγRIIa and FcγRIIb). The number of basophils expressing FcγRIIa and FcγRIIb after one year of VIT significantly increases [57,58,59].

The response of mast cells can be monitored by baseline serum tryptase, which decreases over time of VIT. The tryptase level reflects the mast cell load or their activity. It has been shown that VIT is associated with a small, but a continuous decrease in the baseline serum mast cell tryptase level, indicating a reduced mast cell function [60].

### 5.2. Regulatory Cells of the Immune System

VIT is associated with a progressive increase in peripherally derived Tregs, defined as CD4+ cells, expressing high levels of CD25 and/or FOXP3. An increase in both the proportion and absolute counts of Treg cells subsets in the peripheral blood was observed. These results were positively correlated with the ratio of IgG4/IgE, supporting a role of Tregs in the induction of immune tolerance. Furthermore, the studied patients showed significantly reduced frequencies of CD4+CD25+ and CD4+CD25+FOXP3+ T cells at baseline, which, after six months of VIT, normalized and the monitored levels were similar to those observed in non-allergic individuals [61].

Transcript factor forkhead box protein 3 (Foxp3) is the key factor in the differentiation and function of a subset of Tregs, namely CD4+CD25+. It serves as a vital mechanism of the negative regulation of immune-mediated inflammation [62,63]. *FOXP3* mRNA was increased significantly after 1 year of VIT, further supporting an increase in circulating Tregs [61]. As a result of VIT, Tregs, by producing several cytokines, including IL-10, suppress IgE production and induce allergen-specific IgG4 [61,63,64].

Regulatory B cells (Bregs) are a heterogeneous group of immunosuppressive B cell subsets [65]. Bregs suppress effector T cell responses and induce Tregs differentiation, inhibit dendritic cell maturation, and promote IgG4 production [66,67,68]. The frequency of allergen-specific CD73- CD25+ CD71+ Bregs secreting IL-10 (Br1), with the capacity of suppressing antigen-specific CD4+ T cell proliferation and the ability to produce IgG4, showed a two- to five-fold increase after three to four months of VIT [67,69,70].

### 5.3. Other Immune Cells

Macrophages are cells of the innate immune system capable of phagocytosis, antigen presentation, and cytokine production. They contribute to both pro- and anti-inflammatory processes, thereby influencing immune homeostasis [68]. The existence of several distinct macrophage subsets is well documented. VIT triggers IgG4 production resulting in M2a macrophage subset conversion to M2b-like suppressive macrophages capable of producing IL-10 [71].

### 5.4. Antibodies

The presence of IgE and the titers of sIgE do not predict the severity of clinical reaction, but rather represent the reaction probability [72]. A transient early increase of IgE levels was observed during the first months of allergen immunotherapy, without an increase in allergic symptoms [28,44,72]. Prolonged immunotherapy leads to a decrease in IgE levels [73]. James et al. demonstrated that inhibitory bioactivity, rather than absolute levels of IgE and IgG antibodies, was associated with clinical tolerance of grass pollen immunotherapy [74]. In contrast with grass pollen immunotherapy, the persistent decline in sIgE levels [40], rather than serum inhibitory activity, may play a bigger role in obtaining long-term tolerance in VIT [75,76].

The presence of IgG antibodies is of great significance in the early stages of VIT, but later on, the number of antibodies decline. Even though the levels of IgG progressively reduce, their impact on the inhibition of basophil activation remains [57,58,59].

### 5.5. Cytokines

Certain cytokine dynamics can be observed during VIT, reflecting the mechanisms underlying induced immune tolerance to venom allergens. Two known soluble factors with tolerogenic properties detected during VIT are IL-10 and TGF-β. IL-10 is one of the major cytokines produced by Tregs and Br1 cells, and is involved in the suppression of allergen-specific effector T cells during VIT. IL-10 inhibits the production of total and specific IgE, while increasing IgG4 levels [64]. TGF-β, secreted by many cell types, including Tregs, is a very important cytokine assuring peripheral immune tolerance. Its effect on the effector cells of allergies is complex and diverse [41]. TGF-β induces chemotaxis, reflected in the blocked expression of FcεRI on mast cells [65]. It is also associated with the conversion of naïve CD4+CD25- T cells into functional Tregs [77]. Rather than in subcutaneously administered VIT, it plays a crucial role in oral allergen-specific immunotherapy. Low levels of IL-4 in the build-up phase of VIT rise after 3 weeks of VIT, and fall progressively afterwards. Higher levels of IFNy can be observed after 2 months of VIT. Looking at the dynamics of the described cytokines, a shift from Th2 towards the Th1 response can be seen [49].

## 6. Recurrent Systemic Reaction to a Sting after Stopping VIT

The immune tolerance established during VIT is considered to be lifelong, even after the discontinuation of treatment [17,18,19]. Most studies on the mechanisms and effectiveness data are obtained during VIT. Despite the high success rate of VIT, allergen tolerance may not persist for a prolonged time. Several surveys have been executed since 1985 studying the long-term persistence of tolerance to *Hymenoptera* venom in VIT-treated patients. While 12 surveys covered a period of more than five years, seven extended the time frame of post-VIT observation beyond 10 years, and one focused on a time period of more than 20 years post-therapy [78]. Studies assessed the persistence of tolerance to *Hymenoptera* venom in VIT treated patients by evaluating the reactions after being re-stung. Since 1985, 20 studies on adults have been conducted, interviewing patients from one month to 29 years after stopping VIT. The percentage of SRs for re-sting patients varied from 0% to 28.5% [78]. Bonadonna et al. recently published a study evaluating mastocytosis as a risk factor in VIT, and found that among the 19 patients who were field re-stung after a mean time of 5 years after VIT discontinuation and presented SRs, 18 (94.7%) had mastocytosis and 1 had nonclonal mast cell activation syndrome [21], further supporting the consensus that for patients with mast cell disorders, VIT should be lifelong. 

However, knowledge of the immune mechanisms that persist after discontinuing VIT and assure long-term tolerance is lacking. In a recent study by Adelmeyer et al., levels of *Hymenoptera* venom specific IgE, IgG, and IgG4 were described after 1–29 years of stopping VIT treatment. The results showed similar median concentrations of specific antibodies between re-stung tolerant patients and patients with severe reactions [78].

## 7. Conclusions

Our understanding of the molecular mechanisms underlying allergic diseases and the tolerogenic mechanisms gained during allergen-specific immunotherapy advanced in the last years. Knowledge of the mechanisms that play essential roles in the restoration of the immune system is being joined together to create a complete picture of immune tolerance. Nowadays, it is well accepted that the initial desensitization of effector cells, namely mast cells and basophils, plays a crucial role in preventing anaphylaxis during the build-up phase of VIT, and their decreased reactivity gained during the years of VIT treatment leads to induced long-term immune tolerance. Furthermore, the early generation of blocking IgG antibodies followed by a slow, but constant decrease in IgE levels, assures long-term tolerance. In addition, the generation of Tregs and Bregs and their anti-inflammatory cytokines have been proven to be essential for achieving allergen tolerance. Monitoring the cytokine profile of allergic patients during VIT suggests a shift from Th2 to Th1 immune response. 

Finally, a better understanding of immunological mechanisms occurring during VIT can lead to the development of immune monitoring of venom immunotherapy, determining the end-point by using appropriate biomarkers, and might open new avenues for therapeutic interventions.

## Figures and Tables

**Figure 1 cells-10-01575-f001:**
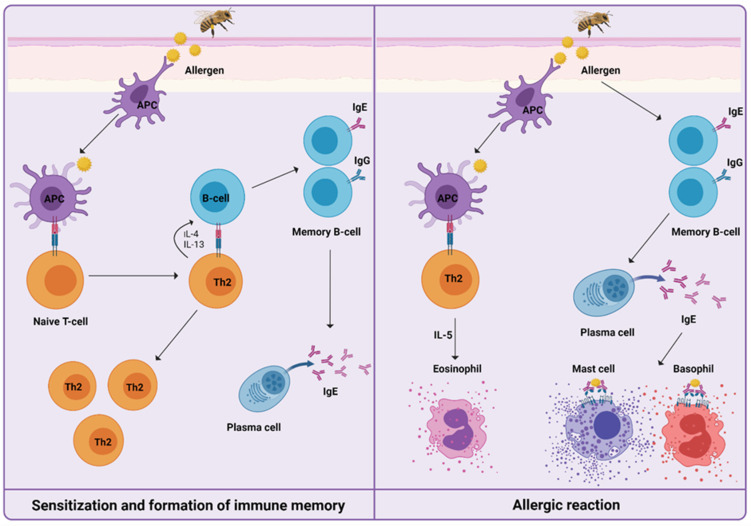
Mechanisms of a *Hymenoptera* venom allergic reaction can be divided into sensitization and formation of immune memory after the first exposure to venom, and the development of type I hypersensitivity reaction upon re-exposure to venom. Only cells and mediators previously described and further involved specifically in venom immunotherapy are shown.

**Figure 2 cells-10-01575-f002:**
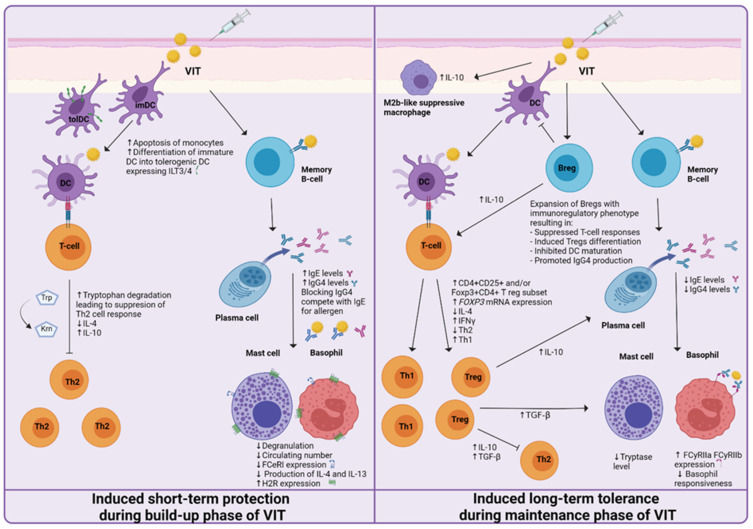
Proposed mechanisms of induced short-term protection and induced long-term tolerance of venom immunotherapy (VIT). The short-term protection effect seems to be established by the generation of blocking IgG antibodies and the desensitization of mast cells and basophils. Induced long-term tolerance is reached after at least three years of VIT. A decrease in basophil responsiveness, shift from Th2 to Th1 immune response, and the generation of regulatory T and B cells has been proven to be essential for achieving allergen tolerance.

**Table 1 cells-10-01575-t001:** Scheme of subcutaneous *Hymenoptera* venom administration according to different protocols [12,23].

	Day 1	Day 2	Day 3	Day 4	Day 8	Day 11	Day 15	Day 22	Day 29	Day 36	Day 43	Day 50
Ultra-rush	0.1–* 100 µg			* 2 × 50 µg		* 100 µg						
Rush	0.01–2 µg	4–20 µg	40–80 µg	* 100 µg	* 100 µg							
Cluster	0.001–0.1 µg				1–10 µg		20–30 µg	2 × 50 µg	* 100 µg	* 100 µg		
Conventional	0.01–0.1 µg				1–2 µg		4–8 µg	10–20 µg	40 µg	60 µg	80 µg	* 100 µg

* Maintenance dose reached.

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
