# Peer review of "Hymenoptera Venom Immunotherapy: Immune Mechanisms of Induced Protection and Tolerance"

_cells, 2021, doi:10.3390/cells10071575_

Round 1
Reviewer 1 Report
This review article is a fairly comprehensive analysis of the mechanisms of venom immunotherapy for bee sting allergy. The one area that is poorly described in the role of regulatory cells in the immune system beginning on line 216. The sentence of line 217 does not make sense. Regulatory T cells are currently divided into 3 subgroups. CD4+CD25+Foxp3+ T cells are generated in the thymus (Treg). CD4+CD25+Foxp3+ T cells can also be generated in peripheral tissues from conventional cells (pTreg). Most importantly so-called Regulatory 1 cells (Tr1) are generated from conventional cells in the periphery and are Foxp3-CD25- but express LAG3. This latter group likely produces IL-10 in response to VIT. The authors should review reference 36 where the roles on the different Treg subtypes are clearly delineated and their role in bee sting allergy defined.
Author Response
Thank you for your careful examination of our manuscript. We have corrected the paragraphs describing the roles of regulatory cells according to the Reviewer's suggestion, both in »Mechanisms of short-term protection« and »Mechanisms of induced long-term tolerance«.
In the revised manuscript the paragraph describing the roles of regulatory cells in short-term protection (lines 159-170) now reads »Maintenance of peripheral tolerance to antigens, including allergens, requires a fine balance between regulatory and effector T cells. Tregs represent a heterogeneous group of T-cell subsets. Briefly, naïve CD3+CD4+ T cells can be classified as thymus-derived CD25+FOXP3+ Tregs or peripherally derived Tregs generated outside the thymus. Peripheral Tregs can be sorted as peripherally induced FOXP3+ T cells, IL-10 producing Tregs (Tr1) and TGF-β producing Th3 cells [39]. Studies suggest that thymus-derived Tregs are specific for self-peptides, whereas peripherally induced Tregs are required to avoid pathologies triggered by antigens [40]. Tregs possess the suppressive capacity acting at different levels of immune mechanisms. There are four main mechanisms of anti-inflammatory activity of Tregs: i) suppressive cytokines (IL-10, TGF-b, IL-35), ii) metabolic disruption mechanisms, iii) suppression of dendritic cell (DC) activation by membrane-bound molecules and iv) cytolysis [39, 41].«
While the paragraph describing the roles of regulatory cells in long-term tolerance (lines 237-244) now reads »VIT is associated with a progressive increase of two types of peripherally derived Tregs defined as CD4+ cells, expressing high levels of CD25+ and/or FOXP3+CD4+ T cells. An increase in both the proportion and absolute counts of Treg cells subsets in the peripheral blood was observed. These results positively correlated with the ratio of IgG4/IgE supporting a role of Tregs in the induction of immune tolerance. Furthermore, studied patients showed significantly reduced frequencies of CD4+CD25+ and CD4+CD25+FOXP3+ T cells at baseline which after six months of VIT normalized and monitored levels were similar to those observed in non-allergic individuals [61].«
Reviewer 2 Report
The manuscript by Demšar Luzar and coauthors entitled “Hymenoptera venom immunotherapy: Immune mechanisms of induced protection and tolerance” reviews recent advances in Hymenoptera venom immunotherapy. The authors describe immune mechanisms during venom immunotherapy as well as mechanisms of short-term and long-term protection. The manuscript is well written and logical. Unfortunately, the authors do not provide the descriptions of different protocols for venom immunotherapy and do not discuss their efficacy. This information would make the manuscript even more interesting and useful for the readers.
Author Response
Thank you for your careful examination and useful suggestions for the improvement of our manuscript. According to the Reviewer suggestions, we have added a paragraph (lines 47-69 of the revised manuscript) as well as a table (Table 1) describing different protocols used in Hymenoptera venom immunotherapy.
The paragraph descibing different protocols of Hymenoptera venom immunotherapy now reads »Immunotherapy aims to restore immune tolerance and thus eliminate systemic allergic reactions after insects’ stings [7]. The first immunotherapy using pure venom extract was carried out in 1974 [8]. Since then, many improvements have been made. Venom immunotherapy is a procedure in which insect venom preparations are administered as a series of subcutaneous injections. It is a two-step procedure consisting of the build-up phase and the maintenance phase [7]. The time to reach the maintenance dose depends on the protocol used; namely conventional, rush, ultra-rush or cluster protocol. The build-up phase can take several weeks or months in conventional protocols [9] or only a few days or hours in a rush or ultra-rush protocols [9, 10]. Cluster protocol represents an alternative regimen for conventional protocol. Recommended starting dose is between 0.001 and 0.1 µg. Subcutaneous injections in the maintenance phase are usually given in four-week intervals in the first year of treatment, every six weeks in the second year of treatment, and every eight weeks from the third to the fifth year of VIT [11]. A maintenance dose of 100 µg is used in the majority of patients. In patients with SR after a field sting or sting challenge while on 100 µg maintenantce dose an up dosing to 200 µg is recommended [7]. The detailed scheme of the particular protocol is shown in Table 1 [7, 12]. The protocol used may be adapted individually regarding patients` reactivity.
In general, Hymenoptera VIT is considered to be safe, although in some cases po-tentially life-threatening SR can occur. It was suggested that rush/ultra-rush protocols can results in a higher rate of SR compared to cluster or conventional protocols. However, the study data addressing this issue are conflicting [13]. In the latest study, Pospischil et al. demonstrated that accelerated VIT protocols, namely rush and ultra-rush, are safer than cluster protocols as they displayed fewer SR [14].«